# Efficacy and Safety of Epidermidibacterium Keratini EPI-7 Derived Postbiotics in Skin Aging: A Prospective Clinical Study

**DOI:** 10.3390/ijms24054634

**Published:** 2023-02-27

**Authors:** Jihee Kim, Young In Lee, Seyoung Mun, Jinuk Jeong, Dong-Geol Lee, Misun Kim, HyungWoo Jo, Sieun Lee, Kyudong Han, Ju Hee Lee

**Affiliations:** 1Department of Dermatology & Cutaneous Biology Research Institute, Yonsei University College of Medicine, Seoul 03722, Republic of Korea; 2Scar Laser and Plastic Surgery Center, Yonsei Cancer Hospital, Seoul 03722, Republic of Korea; 3Department of Dermatology, Yongin Severance Hospital, Yonsei University College of Medicine, Yongin 16995, Republic of Korea; 4Center for Bio Medical Engineering Core Facility, Dankook University, Cheonan 31116, Republic of Korea; 5Department of Bioconvergence Engineering, Dankook University, Yongin 16890, Republic of Korea; 6Department of Microbiology, College of Science & Technology, Dankook University, Cheonan 31116, Republic of Korea; 7R&I Center, COSMAX BTI, Seongnam 13486, Republic of Korea; 8Global Medical Research Center, Seoul 06526, Republic of Korea

**Keywords:** microbiome, actinobacteria, aging, skin, cutibacterium

## Abstract

The present study investigated the effect of topical application of *Epidermidibacterium Keratini* (EPI-7) ferment filtrate, which is a postbiotic product of a novel *actinobacteria,* on skin aging, by performing a prospective randomized split-face clinical study on Asian woman participants. The investigators measured skin biophysical parameters, including skin barrier function, elasticity, and dermal density, and revealed that the application of the EPI-7 ferment filtrate-including test product resulted in significantly higher improvements in barrier function, skin elasticity, and dermal density compared to the placebo group. This study also investigated the influence of EPI-7 ferment filtrate on skin microbiome diversity to access its potential beneficial effects and safety. EPI-7 ferment filtrate increased the abundance of commensal microbes belonging to *Cutibacterium*, *Staphylococcus*, *Corynebacterium*, *Streptococcus*, *Lawsonella*, *Clostridium*, *Rothia*, *Lactobacillus*, and *Prevotella*. The abundance of *Cutibacterium* was significantly increased along with significant changes in *Clostridium* and *Prevotella* abundance. Therefore, EPI-7 postbiotics, which contain the metabolite called orotic acid, ameliorate the skin microbiota linked with the aging phenotype of the skin. This study provides preliminary evidence that postbiotic therapy may affect the signs of skin aging and microbial diversity. To confirm the positive effect of EPI-7 postbiotics and microbial interaction, additional clinical investigations and functional analyses are required.

## 1. Introduction

The diversity and function of the human skin microbiome have recently garnered significant attention in the field of dermatologic research [1,2]. Numerous studies suggest that changes in the cutaneous microbiota have a role in the etiology of a variety of dermatoses [3,4]. Among skin microbiome-related problems, skin aging—characterized by wrinkles, laxity, loss of elasticity, and the appearance of a rough texture [5]—has already been one of the main themes in several metagenomics investigations [6,7,8]. In those studies, multiple comparisons were made between the skin microbiomes of younger (20–30 years old) and older (50–60 years old) women in Western Europe and Asia. The studies have shown that there is a connection between skin microbiomes and the aging process of the skin [6,8,9].

In cosmetic product development, the desire for the discovery of a novel cosmeceutical component with an anti-aging effect stays at a high level [10,11,12]. Recent research has deciphered the relationship between the skin microbiota and skin aging [6,8,9], and only a handful of microbiome-derived antioxidants have been created as cosmeceutical active ingredients. Derived from the fungus *Rhizomucor miehei*, kojic acid is known to have a pigment-lightening effect due to its anti-tyrosinase action [13,14]. Meanwhile, one of the most commonly used bacterial species in this field is *Lactobacillus*. *Lactobacillus* is known for its ability to ferment sugars and produce lactic acid, a highly favored ingredient in skincare products due to its moisturizing and exfoliating properties [15,16]. Additionally, other bacterial species such as *Streptococcus thermophilus* and *Bacillus subtilis* are also utilized for their production of hyaluronic acid, enzymes, and other beneficial compounds [17,18]. Hence, preliminary researches indicate that probiotics, prebiotics, and postbiotics either orally or topically have a positive effect on skin aging [19,20,21].

*Epidermidibacterium Keratini* (EPI-7) is a novel actinobacteria isolated from the skin microbiomes of two different age groups of females, which was found to be more prevalent in the 20s age group [22,23]. EPI-7 ferment filtrate has previously shown its anti-aging property by modulating activities of matrix metalloproteinases (MMPs) in UV-induced photo-aging in human dermal fibroblasts [23]. Among the EPI-7 ferment filtrates, orotic acid is known as a metabolite that plays a crucial role in the biosynthesis of pyrimidines, the nitrogen-containing building blocks of DNA and RNA. Additionally, it is involved in the urea cycle, a metabolic pathway that is responsible for the removal of excess nitrogen from the body [24]. The significance of orotic acid extends beyond its biological functions, as it has garnered attention for its potential therapeutic applications [25]. For example, orotic acid has been used in the treatment of some genetic disorders, such as hereditary orotic aciduria [26] and some forms of urea cycle disorders [24], as it can help improve the body’s ability to produce and excrete urea. However, there is limited research on the use of orotic acid in cosmetics, and its potential benefits and drawbacks have not been well established. Thus, in this study, we aim to provide an overview of the current state of knowledge on Epi-7 orotic acid, its biological and clinical functions, and its potential in the maintenance of skin microbiome diversity.

In the present study, we aimed to assess the efficacy against the signs of skin aging with topically applied EPI-7 ferment filtrate over a 3-week period. This study also investigated the influence of EPI-7 ferment filtrate on skin microbiome diversity to access potential beneficial effects and its safety.

## 2. Results

### 2.1. Participants’ Characteristics

Fifty-five healthy Korean women were enrolled in the 3-week randomized double-blinded prospective clinical study. The numbers of participants included in each age sub-group were as follows: 20 participants (19–20 years of age), 20 participants (50–59 years of age), 15 participants (over 60 years of age). Among these participants, three dropped out from the study due to the poor compliance with the study schedule. The average age of participants was 47.1 ± 13.9 years.

### 2.2. Clinical Efficacy on Skin Hydration and Transepidermal Water Loss (TEWL)

During the study, noninvasive biophysical parameters of the skin, including those measured from Corneometer, Tewameter, Cutometer, Mexamter, and Ultrascan UC22, were measured at baseline and 3 weeks after the initiation of treatment (end of study). The skin hydration measured by Corneometer showed a significant increase from 63.3 ± 9.9 to 70.8 ± 8.3 A.U., before and after the 3-week treatment with the test product; the control side showed a mild increase in skin hydration from 63.0 ± 10.1 to 64.1 ± 10.5 A.U. (*p* < 0.001, paired sample *t*-test, Figure 1A). When comparing the increment in skin hydration between the test and the control groups after 3 weeks of application, the degree of change in the test group was significantly higher in the test side compared to the control-side (11.9%, 1.7% increase, respectively; *p* < 0.001, paired sample *t*-test; Figure 1B). Additionally, the transepidermal water loss (TEWL) was measured on baseline and 3 weeks after the application of the test and the control products. As the result, the test side showed a significant reduction in TEWL after the 3-week study from 16.6 ± 3.8 to 14.5 ± 3.3 g/m^2^h (*p* < 0.001, paired sample *t*-test), while the control side showed no significant difference on baseline and after the end of the study (Figure 1C). The comparison of change in TEWL showed its significant reduction on the test side compared to the control side (−12.5%, −1.7%, respectively; *p* < 0.001, paired *t*-test; Figure 1D).

### 2.3. Clinical Efficacy on Skin Elasticity and Dermal Density

The skin elasticity measured on baseline and after the end of the 3-week study showed a significant increase in the skin surface elasticity (R2) in the test side (*p* < 0.001, paired sample *t*-test), from 0.65 ± 0.07 to 0.69 ± 0.07, while the control side showed no difference (Figure 2A). The comparison in improvement in skin elasticity between the test and the control sides showed a significantly greater improvement in the test group after the end of the 3-week study (5.6%, −0.6%, respectively; *p* < 0.001, paired *t*-test; Figure 2B). The skin density in the test group also increased significant from 61.8 ± 4.9% at baseline to 64.7 ± 4.9% at week 3 (*p* < 0.001, paired sample *t*-test), while the control group showed no significant differences (63.1 ± 5.6%, 62.8 ± 5.5%, respectively; *p* < 0.001, paired *t*-test; Figure 2C). Similarly, the improvement in the dermal density of the test group was significantly higher compared to the control group (4.7%, −0.5%, respectively, *p* < 0.001, paired *t*-test; Figure 2D,E).

### 2.4. Clinical Efficacy on Skin Erythema and Melanin Index

In order to explore the effect of the test product on erythema and pigmentation, additional measurements on the erythema index (E.I.) and melanin index (M.I.) via Mexameter were performed. The test side showed significant reduction in E.I. before and after the end of the study (130.2 ± 31.3, 117.4 ± 28.9, respectively; *p* < 0.001, paired sample *t*-test), while the control side showed no significant difference (Figure 3A). The comparison in the degrees of reduction in E.I. after the 3-week study between the test and the control group also showed a significant difference (*p* < 0.001, paired sample *t*-test; Figure 3B). Meanwhile, the M.I. showed significant decrease in the test group, from 130.2 ± 31.4 to 117.4 ± 28.9, while in the control group it showed only a mild decrease (*p* < 0.001, paired sample *t*-test; Figure 3C). The degree of reduction in the M.I. was also significantly greater in the test side compared to the control side (*p* < 0.001, paired sample *t*-test; Figure 3D).

### 2.5. Effect on Skin Microbiome Diversity

A total 132 swab samples of controls (*n* = 59) and postbiotics applied sites (*n* = 73) were obtained from the study population for sequencing. The 16S metagenomic analysis showed minor differences in microbiota diversity between controls and test sites before the postbiotics (test product) application, as observed by the genus diversity through beta-diversity comparison. Meanwhile, the α diversity comparison between the participants’ subgroups showed statistically significant difference (Kruskal–Wallis test, *p* < 0.05) after 3 weeks application of postbiotics, analyzed by the observed feature diversity index and Chao2 index for richness (Figure 4A; C-0W: control side at week 0, C-3W: control side at week 3, T-0W: test side at week 0, T-3W: test side that week 3). We confirmed from the alpha diversity comparisons that EPI-7 derived postbiotics treatment was closely related to the increase in skin normal microbiota, as the T-3W group showed statistically significant increase in “observed features” and “Chao1” indices, compared to T-0W group. Next, we clustered the estimated microbial compositions for each group among the four comparison sets (C-0W, C-3W, T-0W, T-3W) through beta-diversity analysis. Distance matrix analysis from principal coordinates analysis (PCoA) of unweighted UniFrac phylogenetic distances showed significant change in β diversity on the postbiotics applied site after three weeks (T-3W) (Figure 4B). As a result, we confirmed that the microbial compositions of each sample were clearly divided by EPI-7 postbiotics treatment in T-3W. In addition, it has been confirmed that microbial cluster changes in the intergroup were evident, depending on whether the probiotic product was treated or not.

Analysis by 16S metagenomics for comparison of differential abundance between each group revealed that Cutibacterium and Staphylococcus were the most abundant within all groups. The top 10 skin resident microbes were analyzed and the relative microbial frequency included Cutibacterium, Staphylococcus, uncultured, Corynebacterium, Streptococcus, Lawsonella, Clostridium, Rothia, Lactobacillus, and Prevotella, which are well-known skin commensals within the published literature, indicating the versatility of our 16S metagenomic data (Figure 5A). We observed a clear and statistically significant change in Clostridium and Prevotella abundance after 3-week application of postbiotics (Figure 5B).

### 2.6. Safety Profile

No adverse effects, including application-site itching, pain, or post-treatment erythema were observed during the 3-week period of study in the test group. None of the participants dropped out of the study because of adverse events, suggesting that the topical formulation of the test material was safe to use.

## 3. Discussion

In this study, we investigated changes in skin biophysical parameters and microbiome in healthy subjects after 3 weeks of EPI-7 derived postbiotics application. Our results demonstrated that the topical application of the postbiotics substantially enhanced the skin barrier function as assessed by corneometer and TEWL, possibly due to the restoration of commensal microbial diversity. In addition, skin elasticity and dermal density were also significantly improved. Meanwhile, the metagenomics analysis revealed that the use of the postbiotics substantially altered the diversity of the skin microbiome, which included *Cutibacterium*, *Staphylococcus*, *Corynebacterium*, *Streptococcus*, *Lawsonella*, *Clostridium*, *Rothia*, *Lactobacillus*, and *Prevotella*. In particular, the abundance of *Clostridium* and *Prevotella* were significantly changed.

When describing metabolomics analysis, alpha diversity is a measure of the species diversity within a single ecosystem or a sample group. It provides information on the number of species and their relative abundance in a given area including the skin. Here, we considered two aspects when predicting alpha diversity in species. “Species richness” and “evenness” are the two components of alpha diversity that describe microbial diversity within a community. Species richness refers to the number of different species present in a community. It provides a simple count of the number of species within a sample group, thereby giving an idea of the overall number of different species present. Species evenness, on the other hand, refers to the distribution of individuals among species within a community. It measures how evenly the individuals are distributed among species within a sample group [27].

In this study, the alpha diversity results based on “observed features” and “Chao1” indices considering the “richness” of microbial genus suggested that the topical application of EPI-7 clearly increased the normal microbial diversity after 3 weeks. Meanwhile, no statistically significant changes were found in the alpha diversity analysis considering the microbial “evenness” among each study group (Shannon, Simpson, and Pielou’s evenness). Therefore, we hypothesized that the EPI-7 postbiotics, including orotic acid treatment, did not significantly alter the dominance of a specific microorganism in the skin microbiome, but rather played a crucial role in enhancing the normally-distributed diversity of microbial species within the skin environment.

Skin aging is the result of variations in the cutaneous structure and physiological changes [5], accompanied by a decrease in species diversity and changes in the predominant bacteria [6,8,9]. Postbiotics are byproducts of probiotic bacteria responsible for their primary biological functions. Typically, postbiotics are produced by microbial metabolism in a fermented matrix [28,29]. They are feasible to incorporate into products, in contrast to probiotics, which provide formulation and packaging challenges to ensure the survivability of the bacteria [28,29]. In recent years, several studies on the use of postbiotics have been performed in aesthetic medicine and skin disease [29,30,31].

In a study involving 20 healthy women over 60, a topical cream containing a probiotics lysate of the bacterium *Streptococcus thermophiles* was applied for 15 days. The postbiotics used in the study contained sphingomyelinase, which has the potential to induce ceramide production in the skin, which was indicated by increased hydration in the stratum corneum [32,33]. Additionally, *Staphylococcus epidermidis,* a commensal bacteria, is known to ferment the glycerol, a significant component in stratum corneum, into butyric acid [30,34]. The application of butyric acid showed in vivo anti-inflammatory effects in UV-induced skin inflammation [29,34].

Meanwhile, another previous study aimed to evaluate the moisturizing effect of a skin moisturizer containing paraprobiotics (heat-inactivated Bifidobacterium lactis and Lactobacillus plantarum) and its impact on the skin microbiome of healthy individuals. Fifty subjects were divided into a treatment group (25 individuals) and a control group (25 individuals), and the changes in transepidermal water loss (TEWL) and stratum corneum moisture (SCM) after 4 weeks of treatment were observed [35]. As the result, the investigators observed skin microbiome changes, including the increase in the numbers of common microorganisms belonging to the phylum *Planctomyces*, *Chloroflexi*, *Verrucomicrobia*, *Gemmatimonadates*, *Planctomyces*, and *Nitrospirae*. In addition, significantly improved TE and SCM were also noticed. Interestingly, the investigators also emphasized that the microbial diversity richness was significantly increased after treatment of paraprobiotics, similar to the results of our current study.

In summary, this study highlighted the effect of EPI-7 derived postbiotics on biodiversity and microbial composition on the faces of healthy women. It can be speculated that microbial change reflects changes in skin physiology and appearance. Limitations include the size of the study population and short study period. Consequently, the present investigation provides early evidence indicating how postbiotic treatment may influence the phenotypes of skin aging and microbial diversity. Further thorough research, including an examination of postbiotic metabolites, is required to validate the results.

## 4. Materials and Methods

### 4.1. Clinical Study

#### 4.1.1. Ethics Statements

This clinical trial was approved by the Institutional Review Board (IRB) of Global Medical Research Center (IRB numbers: GMRC-22425-EA1). The study protocol was initiated after written informed consent was obtained from all participants, and the study adhered to the principles of the Declaration of Helsinki.

#### 4.1.2. Study Participants

This study was performed from April 2022 to June 2022. Healthy adult women, aged 19–69 years, were eligible for this study. The main exclusion criteria were as follows: major internal disorders, such as cancers, major or active skin diseases, current pregnancy or lactation, history of allergy or hypersensitivity reactions, history of undergoing dermatologic procedures (e.g., lasers, filler, botulinum toxin injection) in the past 3 months, use of topical medications with any skin reactions (e.g., glucocorticoids, retinoids, and topical immunomodulators) in the past 3 months, use of systemic medications, such as glucocorticoids and immunomodulators within the last 1 month, and application of skin care products or topical medications with similar functions in anti-aging within the last 3 months before participating in the study.

#### 4.1.3. Study Design and Test Product Information

The study was designed as a double-blind, randomized, split-face, vehicle-controlled study. The EPI-7 medium derivative was derived from a previously studied natural product extraction assay using a thin layer chromatography (TLC) separation method [23]. To extract EPI-7 ferment postbiotic candidates, we secured specific metabolites, including orotic acid, that are expressed through the metabolic process after culture through a comparative analysis of EPI-7 before and after bacterial culture with R2A medium (50 L). Of EPI-7 ferment filtrates, we conducted an in-depth study on skin clinical improvement and microbiome changes according to the addition of orotic acid (10 ppm) in this study. The rest material of EPI-7 postbiotic candidates was not disclosed due to business and research interests. All participants topically applied the study product (EPI-7 culture medium derivatives) or the vehicle cream (without the active ingredient) to the randomly selected sides of their face. The active ingredient of the test product contained *Epidermidibacterium Keratini* ferment filtrate, along with 1,2-Hexanediol, Tromethamine, Ethylhexylglycerin, and Carbomer. The vehicle cream was manufactured with identical ingredients with the test product, except for the EPI-7 ferment filtrate. A random number generator (Microsoft Excel 2013 version; Microsoft) assigned application of the test product on either the left or the right side of the face. The subjects applied the vehicle cream as the control product to the non-allocated side of their faces. The subjects applied each product to the designated sides of their faces twice daily for the 3-week study period. Throughout the study, both the investigators and the subjects were blinded to the products applied to each side of the face. The study cream and the vehicle cream were similarly matched in their colors and fragrances, and they were prepared in identical containers, to ensure the double-blinded design of the study.

#### 4.1.4. Assessment of the Clinical Efficacy of the Test Product

The clinical efficacy of the topical test product facial skin rejuvenation was assessed at baseline and 3 weeks after the initiation of treatment. The skin barrier function and skin hydration were evaluated using Corneometer^®^ and Tewemeter^®^ (Courage Khazaka Electronic GmbH, Köln, Germany). Skin elasticity was measured using Cutometer Dual MPA580 (Courage Khazaka Electronics, Köln, Germany), while skin density was additionally measured using Ultrascan^®^ UC22 (Courage Khazaka electronic, Germany), a device that measured the density of the skin by generating a short electric pulse using a 22 MHz ultra-sonic transducer. Lastly, the melanin index and erythema index of the skin were measured using the Mexameter MX18 (Courage Khazaka Electronic GmbH, Köln, Germany). Each measurement was performed thrice during each visit, and we instructed all participants to acclimatize to the controlled environmental conditions for 30 min at each visit (room temperature: 18–21 °C; relative humidity: 40–60%, without direct light), before the measurement.

#### 4.1.5. Assessment of the Safety of the Device

At every visit, patients underwent a physical examination to assess safety outcomes. Patients also were asked to report side effects, such as erythema, swelling, blistering, hyperpigmentation, or hypopigmentation, during and after treatment. The investigators categorized the degree of erythema and burning into four grades: none, mild, moderate, or severe.

### 4.2. Microbiome Characterization

#### 4.2.1. Microbiome Samples Collection from the Study Participants

A total of 132 skin microbial swab samples were collected by dividing two sites (control and postbiotic treatment sites) from the faces of healthy Korean women using the NBgen-GUT NP self-collection tube (Noble Biosciences, Republic of Korea). The participants applied the postbiotics-containing test product or the control product on the randomly assigned sides of the entire face. The microbiome swab was collected in an area of 4 cm x 5 cm on each side of the face, starting 1 cm below the lowest point of the lower eyelid, and 1 cm next to the nasal ala. All collected clinical samples were stored at −80 °C before being used in further steps. The clinical specimen collection in this study (skin microbiota samples collection) was approved by the ethics committee of Global Medical Research Center (IRB numbers: GMRC-22425-EA1). All clinical processing applied in this study was conducted under the guidelines and regulations of the Helsinki Declaration.

#### 4.2.2. Bacterial Genomic DNA Isolation

Bacterial genomic DNA (gDNA) from 132 facial skin microbial swab samples were extracted using the QIAamp PowerFecal Pro DNA Kit (Qiagen, Hilden, Germany), and all experimental processes were performed following the optimal protocols provided by the DNA extraction kit. The quality check of all extracted bacterial gDNA was conducted using the NanoDrop One (ThermoFisher Scientific, Waltham, MA, USA) equipment. All extracted gDNA samples were stored at 4 °C until the following process.

#### 4.2.3. 16S V3-V4 Metagenome Sequencing

The 16S V3-V4 amplicon libraries for Next Generation Sequencing (NGS) based metagenome sequencing were prepared on the recommendation of the official Illumina 16S V3-V4 metagenome sequencing library preparation protocol, which targeted the V3-V4 hypervariable region within the bacterial 16S ribosomal RNA gene (Illumina, San Diego, CA, USA). All PCR amplification for sequencing library preparation was conducted using 2X KAPA HiFi Hot Start Ready Mix (Roche, Charles Avenue, UK). For this experimental purpose, the 16S V3-V4 hypervariable region-specific universal primer pair was used. The primer sequences were as follows:

16S 341F forward primer is 5′ -TCGTCGGCAGCGTCAGATGTGTATAAGAGACAGCCTACGGGNGGCWGCAG-3′. 16S 806R reverse primer is 5′ -GTCTCGTGGGCTCGGAGATGTGTATAAGAGACAGGACTACHVGGGTATCTAATCC-3′. After PCR amplification, all amplicon products were purified using the AMPure XP beads (Beckman Coulter, Brea, CA, USA). The additional PCR amplification was then performed to add the Illumina adapter and multiplex indices using the Nextera XD Index (Illumina, USA). The final PCR products were then purified once again using the AMPure XP beads. After the metagenome sequencing library preparation, the 16S V3-V4 metagenome sequencing was carried out using Illumina Miseq (2 × 300) paired-end sequencing workflow (Illumina, USA).

#### 4.2.4. 16S Microbiome Data Analysis

We analyzed the paired-end 16S V3-V4 raw sequencing read files (i.e., fastq.gz files) using two bioinformatics pipelines: QIIME2 and DADA2 (Version 1.2.0). The executing software for QIIME2 was Python software and for DADA2 was R (version 3.3.2). First, the sequencing reads were sorted using unique barcodes for each PCR product. The barcode, linker, and primer sequences were then removed from the original sequencing reads. The removed reads were merged paired-end reads using FLASH v 1.2.11. The merged reads containing two or more ambiguous nucleotides, those with a low-quality score (average score < 20), or reads shorter than 300 bp, were filtered out. All pre-filtered sequence data quality control and feature table construction were completed using the Divisive Amplicon Denoising Algorithm 2 (DADA2) in the QIIME 2 plugin, which detects and corrects amplicon errors and filters out PhiX chimeric sequences. The pre-processed reads from each sample were used to calculate the number of Amplicon Sequence Variants (ASVs). The number of ASVs was determined by clustering the sequences from each sample using a 99% sequence identity cut-off using DADA2 pipeline-based QIIME2 software. After the denoising step, the microbial diversity analysis (alpha and beta) was performed using the “diversity” category in the QIIME2 plugin. The alpha diversity was calculated by Observed_ASVs, Chao1, Shannon, Simpson, and Pielou_e alpha diversity indices, which were used to estimate microbial richness and evenness scores, and the Kruskal–Wallis nonparametric ANOVA test and the Mann–Whitney U test were applied to determine the statistical significance of each comparison analysis in the present skin microbiome study. Microbial beta-diversity between each comparative group was confirmed through distance dissimilarity calculated using Bray–Curtis and unweighted_UniFrac distance matrices. The permutational multivariate analysis of variance (PERMANOVA) non-parametric statistical test was used to determine the statistical significance of the beta-diversity analysis. The results of the microbial beta-diversity analysis were visualized through a principal coordinate analysis (PCoA) plot using R studio. Finally, taxonomic relative abundance was counted with a naïve Bayes classifier using a confidence threshold of 0.7%, derived from the ASVs data for each sample. The classified microbial composition based on SILVA v138 16S rRNA gene database was normalized using the value calculated from the taxonomy abundance count, divided by the number of pre-processed reads for each sample.

### 4.3. Statistical Analysis

Data within the group were compared using the paired Student’s *t*-test and are presented as the mean ± standard error of the mean. Data were analyzed for normality using the Kolmogorov–Smirnov test. Statistical significance was set at *p* < 0.05. All statistical analyses were performed using SPSS, version 25.0 (IBM Corp., Armonk, NY, USA).

## Figures and Tables

**Figure 1 ijms-24-04634-f001:**
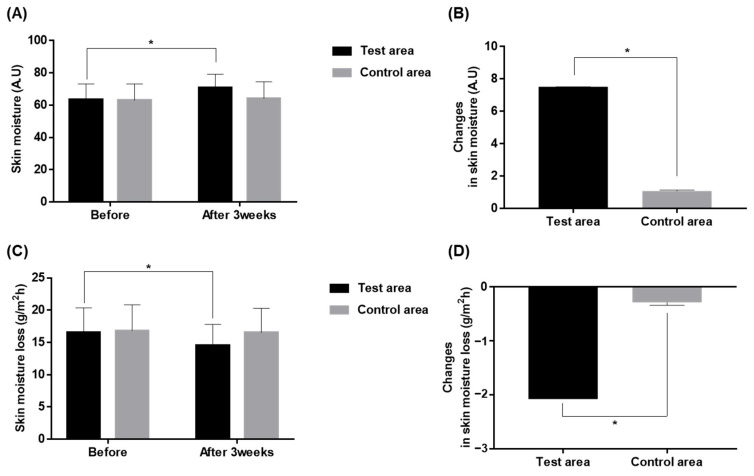
Change in skin moisture (**A**,**B**) and TEWL (**C**,**D**) after 3-week application of the test and the control products, compared to the baseline. * *p* < 0.001, paired sample *t*-test.

**Figure 2 ijms-24-04634-f002:**
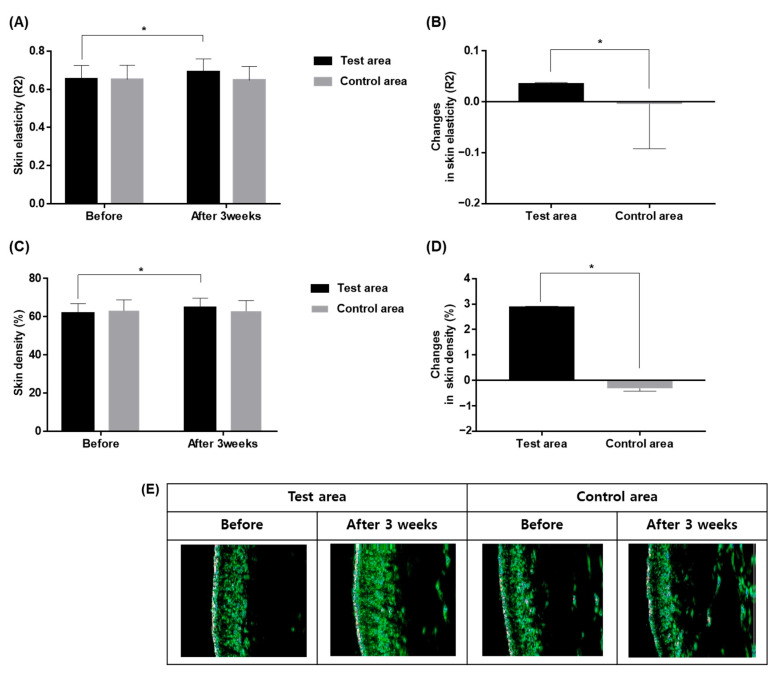
Change in skin elasticity (**A**,**B**) and epidermal density (**C**,**D**) after 3-week application of the test and the control products, compared to the baseline. * *p* < 0.001, paired sample *t*-test.

**Figure 3 ijms-24-04634-f003:**
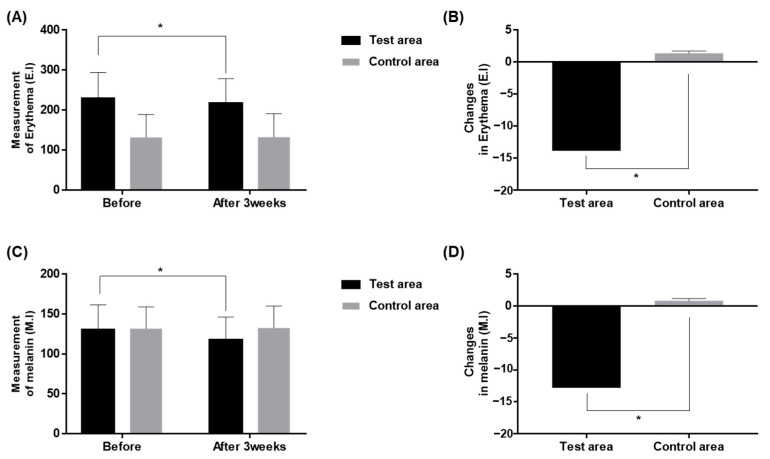
Change in skin erythema (**A**,**B**) and melanin index (**C**,**D**) after 3-week application of the test and the control products, compared to the baseline. * *p* < 0.001, paired sample *t*-test.

**Figure 4 ijms-24-04634-f004:**
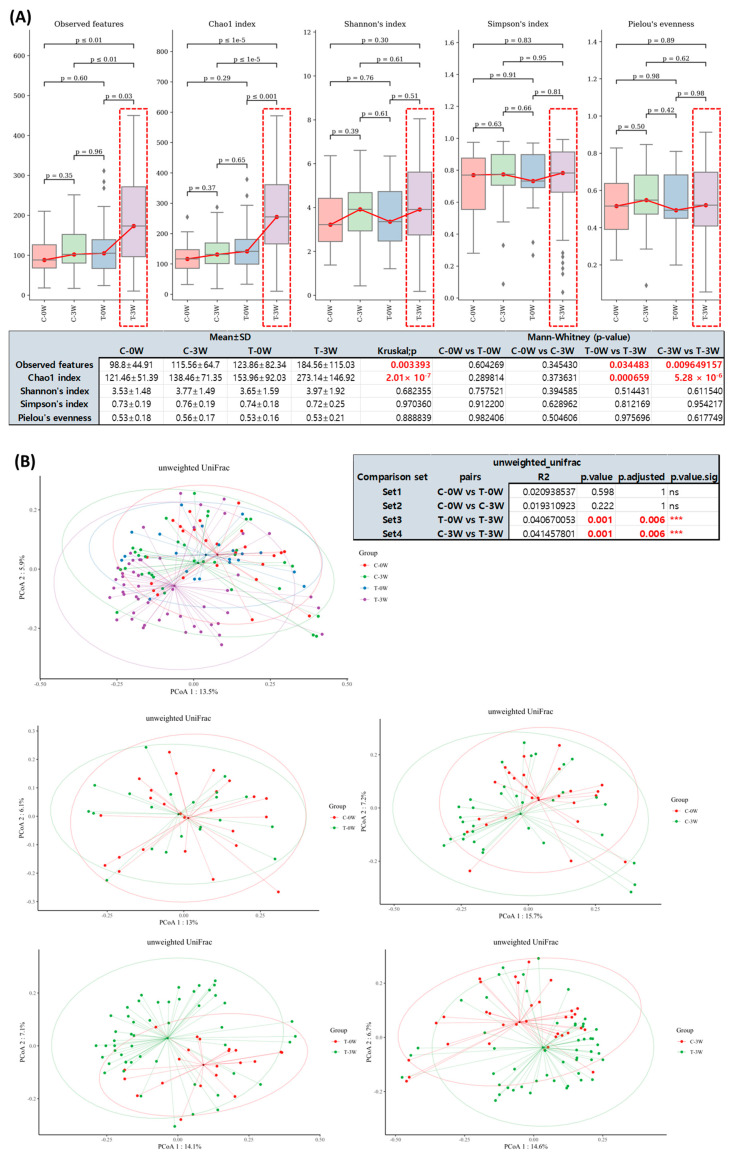
The structure of skin microbiota before and after the postbiotics application. (**A**) Alpha diversity comparison between the participants’ subgroups showing statistically significant differences after 3 weeks application of postbiotics, analyzed by observed feature diversity index and Chao2 index (Kruskal Wallis *p* < 0.05). (**B**) Principal coordinates analysis (PCoA) plot with different relative abundances of OTUS in participants before and after application of postbiotics. C-0W, control 0 weeks; C-3W, control 3 weeks; T-0W, postbiotics applied site 0 weeks; T-3W; postbiotics applied site 3 weeks; OTU, operational taxonomic units. (#Set1, 2, 3, and 4, respectively: Mann–Whitney *p*-value = 0.598, 0.222, 0.001, and 0.001, respectively.) The asterisk (*) represents the *p* value of the statistical test (*** < 0.0001).

**Figure 5 ijms-24-04634-f005:**
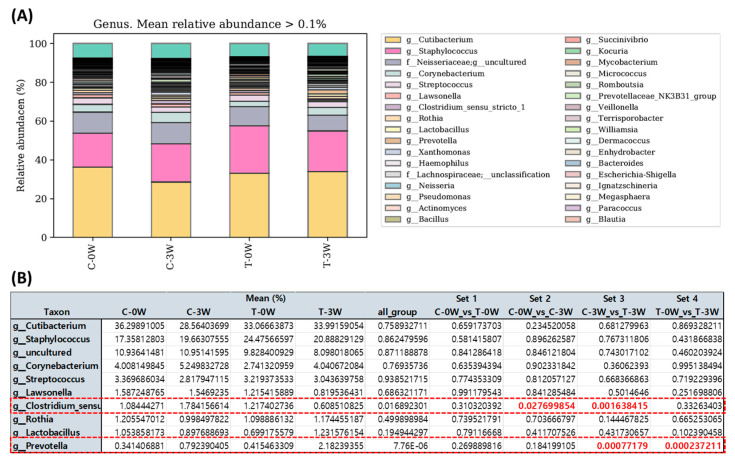
Relative microbial abundance within the participants. (**A**) Relative abundance of skin microbes within each group. (**B**) Changes of relative abundance of top ten microbes before and after postbiotics application. C-0W, control 0 weeks; C-3W, control 3 weeks; T-0W, postbiotics applied site 0 weeks; T-3W; postbiotics applied site 3 week. The red dashed boxes signify a statistically significant decrease in *Clostridium* abundance and an increase in *Prevotella* abundance after three weeks of postbiotic use.

## Data Availability

Data are available from the corresponding author upon reasonable request owing to privacy and ethical restrictions.

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
