# Peer review of "Efficacy and Safety of Epidermidibacterium Keratini EPI-7 Derived Postbiotics in Skin Aging: A Prospective Clinical Study"

_ijms, 2023, doi:10.3390/ijms24054634_

Round 1
Reviewer 1 Report
I most of all miss the scientific information on the origin, production of the EPI-7 extract and use level in the test formulation. I downloaded the manuscript with many notes and suggested corrections. I hope you can see the notes in the manuscript. Why did you treat for 3 weeks? Many epidermal studies are treated for 4 weeks, the time for a complete epidermal turnover. The mechanism of action of the postbiotic extract on dermal aging parameters is not clear.

Author Response
Reviewer1 Comments and Suggestions for Authors
Comment 1: I most of all miss the scientific information on the origin, production of the EPI-7 extract and use level in the test formulation. I downloaded the manuscript with many notes and suggested corrections. I hope you can see the notes in the manuscript.
Response: Thank you for your kind review of our manuscript. We have thoroughly reviewed your comments from our manuscript file and added answers to each comment in the word file, as we attached in the re-submission process (Please check manuscript_with highligts word file).
Comment 2: Why did you treat for 3 weeks? Many epidermal studies are treated for 4 weeks, the time for a complete epidermal turnover.
Response: Thank you for your comment. We sincerely agree with your comment. It had been better to process a 4-week design clinical study, but since the time when we performed the clinical study was mid-May, we wanted to finish the clinical study before the beginning of summer (due to COVID measure, Koreans all wear masks and it can significantly affect facial microbiome especially during the summer season).
Comment 3: The mechanism of action of the postbiotic extract on dermal aging parameters is not clear.
Response: Thank you for your comment. We have further described the previous literatures showing the use of postbiotics extract for anti-aging properties as shown in lines 245-264. Further in vitro studies should be performed in order to understand the molecular pathogenesis on the mechanism of postbiotic extract and its effects on skin aging.

Reviewer 2 Report
The manuscript "EFFICACY AND SAFETY OF AN EPIDERMIDIBACTERIUM KERATINI EPI-7 3 DERIVED POSTBIOTICS IN SKIN AGING: A PROSPECTIVE CLINICAL STUDY" presents a useful research on important factors related with the novel treatment of an Epidermidibacterium Keratini EPI-7 3 derived postbiotics for skin aging. I consider the manuscript valuable for further work with patients/clients in practice, especially in cosmetic dermatology and pharmacy. However, there are some suggestions.
INTRODUCTION: Is there any previous article on this postbiotic use? If not, you may emphasize this data. If yes, you may mention their results from previous articles (e.g. in the Discussion section).
METHODS: It is written „All participants topically applied the study product [EPI-7 culture medium deriv- 236 atives] or the vehicle cream (without the active ingredient) to the randomly selected sides 237 of their face“. Could you give mora data on specific facial localization and skin areas (size) of application? Were the patients dermatologically examined? How did you collect the participants?
RESULTS: In the analysis and throughout the paper, a complex statistical processing was done which gave complex and impressive statistical results. So, the obtained data should be emphasized and further interpreted in Discussion, in order to obtain appropriate conclusions. However, since I am a dermatologist, I can comment on the text as a clinician and I think that it would be useful to check the obtained data by a molecular biologist.
In DISCUSSION: So, I think that Discussion should be expanded and should include more explanations important for the practice. Also, more comparisons with other results of other authors are needed, especially newer results of other authors should be added and commented on.
There are some typos on some parts: for example showd and appilication.
Author Response
Reviewer2 Comments and Suggestions for Authors
Overall comment: The manuscript "EFFICACY AND SAFETY OF AN EPIDERMIDIBACTERIUM KERATINI EPI-7 3 DERIVED POSTBIOTICS IN SKIN AGING: A PROSPECTIVE CLINICAL STUDY" presents a useful research on important factors related with the novel treatment of an Epidermidibacterium Keratini EPI-7 3 derived postbiotics for skin aging. I consider the manuscript valuable for further work with patients/clients in practice, especially in cosmetic dermatology and pharmacy. However, there are some suggestions.
Comment 1: INTRODUCTION: Is there any previous article on this postbiotic use? If not, you may emphasize this data. If yes, you may mention their results from previous articles (e.g. in the Discussion section).
Response 1: We thank for your suggestion. As you recommended, we have added the detailed information on EPI-7 feature and postbiotic function in therapeutic application. And further, the particular cases suggesting a positive effect on skin clearance are also mentioned. [Lines 57-82]
Comment 2: METHODS: It is written „All participants topically applied the study product [EPI-7 culture medium derivatives] or the vehicle cream (without the active ingredient) to the randomly selected sides 237 of their face“. Could you give mora data on specific facial localization and skin areas (size) of application? Were the patients dermatologically examined? How did you collect the participants?
Response 2: Thank you for your comments. Board-certified dermatologists performed initial screening of the participants according to the inclusion and the exclusion criteria given. The participants applied the postbiotics-containing test product or the control product on the randomly assigned sides of the entire face. The microbiome swab was collected in an area of 4cm x 5cm on each side of the face, starting 1cm below the lowest point of the lower eyelid, and 1cm next to the nasal ala. [Lines 346-350]
Comment 3: RESULTS: In the analysis and throughout the paper, a complex statistical processing was done which gave complex and impressive statistical results. So, the obtained data should be emphasized and further interpreted in Discussion, in order to obtain appropriate conclusions. However, since I am a dermatologist, I can comment on the text as a clinician and I think that it would be useful to check the obtained data by a molecular biologist.
Response 3: It is always valuable to consider different perspectives and expertise when analyzing complex data. We agree that seeking the input of a molecular biologist could further enhance the interpretation and understanding of the obtained results. We have added a detailed description of statistical results and their meaning in our study. Please refer to the word file with manuscript_with highlights below.
Comment 4: In DISCUSSION: So, I think that Discussion should be expanded and should include more explanations important for the practice. Also, more comparisons with other results of other authors are needed, especially newer results of other authors should be added and commented on.
Response 3: Thank you for your comment. I agree with you that expanding the Discussion section and adding more explanations and comparisons with other relevant studies would provide a more comprehensive understanding of the results. We have added a detailed discussion based on the previous study. [Lines 214-263].

Round 2
Reviewer 1 Report
answers, corrections and amendments after first review are acceptable